# Efficient and Programmable Exploration of Synthesizable Chemical Space

**Shitong Luo** *luost@mit.edu*
*Department of Electrical Engineering and Computer Science*
*Massachusetts Institute of Technology*

**Connor W. Coley** *ccoley@mit.edu*
*Department of Chemical Engineering*
*Department of Electrical Engineering and Computer Science*
*Massachusetts Institute of Technology*

**Reviewed on OpenReview:** *https://openreview.net/forum?id=xDlIer2UnI*

## Abstract

The constrained nature of synthesizable chemical space poses a significant challenge for sampling molecules that are both synthetically accessible and possess desired properties. In this work, we present PrexSyn, an efficient and programmable model for molecular discovery within synthesizable chemical space. PrexSyn is based on a decoder-only transformer trained on a billion-scale datastream of synthesizable pathways paired with molecular properties, enabled by a real-time, high-throughput C++-based data generation engine. The large-scale training data allows PrexSyn to reconstruct the synthesizable chemical space nearly perfectly at a high inference speed and learn the association between properties and synthesizable molecules. Based on its learned property-pathway mappings, PrexSyn can generate synthesizable molecules that satisfy not only single-property conditions but also composite property queries joined by logical operators, thereby allowing users to "program" generation objectives. Moreover, by exploiting this property-based querying capability, PrexSyn can efficiently optimize molecules against black-box oracle functions via iterative query refinement, achieving higher sampling efficiency than even synthesis-agnostic baselines, making PrexSyn a powerful general-purpose molecular optimization tool. Overall, PrexSyn pushes the frontier of synthesizable molecular design by setting a new state of the art in synthesizable chemical space coverage, molecular sampling efficiency, and inference speed.

**Code:** `https://github.com/luost26/prexsyn`

## 1 Introduction

Advances in generative models have led to the proliferation of new methods for molecular design, offering higher sampling efficiency than enumerative virtual screening (Gómez-Bombarelli et al., 2018; Gao et al., 2022). Predominantly, generative models represent molecules in the form of strings, graphs, or 3D coordinates, which are agnostic to synthesizability. As a result, these models tend to propose molecules that are difficult or even impossible to synthesize in practice when used for molecular optimization (Gao & Coley, 2020). The lack of synthetic tractability has been a major bottleneck hindering experimental validation and translation to biomedical applications.

Methods that focus on generating synthetic pathways rather than unconstrained molecular graphs have emerged as a promising approach to addressing the synthesizability issue (Gao et al., 2021; Swanson et al., 2024; Horwood & Noutahi, 2020; Cretu et al., 2024), since synthesis pathways composed of vendor-validated building blocks and reaction protocols can be executed with an estimated success rate of around 85% (Grygorenko et al., 2020). Among these methods, one notable category is the "chemical space projection" ap-

proach which aims to find structurally similar molecules, or analogs, within the synthesizable space for any molecular graph (Luo et al., 2024). In this paradigm, an external generative model is first used to draft molecular graphs that satisfy specific properties but are not necessarily synthesizable. Then, these molecular graphs are "projected" to synthesizable analogs in form of *postfix notation of synthesis*, a linear and synthetic pathway-based molecular representation.

However, methods in this category face two major challenges. First, their coverage of the synthesizable chemical space remains limited. Even the recent state-of-the-art models (Gao et al., 2025; Sun et al., 2025) reconstruct only about 70% of the Enamine REAL space, where the molecules are derived from the same Enamine building block library used for training. Some other recent methods have reported higher reconstruction rates (Lee et al., 2025), but these improvements come at the cost of substantially increased sampling time due to extensive inference-time search, limiting their practicality for real-world, large-scale applications. Second, the two-stage generation-projection process inevitably creates structural and functional inconsistencies between the molecular graphs and the synthesizable analogs, often leading to degraded or inconsistent properties. These challenges highlight the need to improve coverage of synthesizable chemical space so that models develop better knowledge of the synthesizable landscape. On top of this, it is also important to move beyond graph-conditioned projection toward property-conditioned generation in order to bridge the gap between molecular properties and synthesizability.

Motivated by these challenges, our goal is to develop a generative model that (1) guarantees synthesizability according to predefined reaction rules, (2) has high coverage of the synthesizable chemical space, (3) generates molecules directly within the synthesizable chemical space according to molecular properties, (4) has high inference speed, and (5) has high sample efficiency when used for black-box optimization.

We present **PrexSyn** as a solution. PrexSyn uses the postfix notation of a synthetic pathway as its molecular representation which ensures synthesizability subject to well-defined reaction rules and building blocks (Luo et al., 2024). Moving beyond the previous graph-conditioned projection frameworks (Luo et al., 2024; Gao et al., 2025; Lee et al., 2025), PrexSyn is formulated as a general *property-conditioned* model, which is built upon a decoder-only transformer that autoregressively generates the postfix notations of synthesis conditioned on property prompts. Further, PrexSyn is trained on a billion-scale datastream of synthesizable pathways paired with molecular properties using only two GPUs and 32 CPU cores in two days, which is made possible by a real-time, high-throughput C++-based data generation engine. This represents a substantially larger scale of training while still requiring less compute than prior work.

PrexSyn achieves a record-high 94% reconstruction rate on the Enamine REAL space, surpassing previous methods by a large margin, demonstrating a near-perfect coverage of this synthesizable chemical space. PrexSyn is also significantly faster than any prior methods, achieving over 60 times speedup in inference time compared to the highest-accuracy baseline, enabled by its stronger model capacity that eliminates the need for inference-time search.

We showcase the property-conditioned generation capability of PrexSyn by training it on a set of molecular descriptors widely used in QSAR modeling (Roy et al., 2015), including molecular fingerprints, fragment structures, and multiple physicochemical descriptors. To further enable logical composition of multiple properties connected by logical operators (AND, NOT, OR) (Du et al., 2020), we present a sampling algorithm that compiles logical queries into arithmetic combinations of probability distributions conditioned on each individual property prompt, allowing users to "program" complex generation objectives.

The property-based querying capability of PrexSyn creates a new way of sampling molecules with respect to black-box oracle functions, referred to as *query-space optimization*, where property queries are iteratively refined using oracle feedback. PrexSyn achieves higher sampling efficiency than not only synthesis-based methods but also synthesis-agnostic baselines. This advantage arises because the query space is numerical and therefore well-structured, making it easier to optimize over than the discrete and sparse search spaces of molecular graphs (Jensen, 2019; Gao et al., 2025) or synthetic trees (Swanson et al., 2024; Cretu et al., 2024). The high molecular sampling efficiency, combined with guaranteed synthesizability, makes PrexSyn a powerful general-purpose tool for molecular design and optimization.

## 2 Method

PrexSyn is a framework for synthesizable molecular design that introduces new contributions spanning model architecture, data infrastructure, and inference algorithms. In this section, we will first describe the decoder-only transformer architecture of PrexSyn (Section 2.1), which unifies molecular properties and synthesizable molecular generation. We then present the real-time, high-throughput data generation engine (Section 2.2), which plays the key role in scaling up training to billions of pathways under a modest compute budget. Finally, we detail the inference-time algorithms for sampling molecules under multiple-property constraints expressed as logical queries (Section 2.4), and we show how this querying capability enables efficient molecular sampling against black-box oracles (Section 2.5), making PrexSyn a highly efficient general-purpose tool for molecular design.

### 2.1 Decoder-only transformer architecture unifies property and synthesis

PrexSyn is based on a decoder-only transformer (Vaswani et al., 2017) which takes as input a prompt of molecular properties and then autoregressively generates postfix notations of synthesis (Figure 1b). Formally, the model learns the following conditional distribution:

$$p(\boldsymbol{s}|\boldsymbol{C}) = \prod_{i=1}^{N} p(s_i|s_{<i}, \boldsymbol{C}). \tag{1}$$

Here, $\boldsymbol{C}$ is the property prompt generated by embedding property values with simple linear layers. $\boldsymbol{s} = [s_1, \ldots, s_N]$ is the tokenized postfix notation sequence, where each building block and each reaction corresponds to a unique token associated with a learnable embedding vector. Positional encodings are added to the token embeddings to indicate the order (Vaswani et al., 2017). To predict the next token, the model adopts a two-level approach. First, it predicts the type of the next token (*i.e.*, building block, reaction, [START], or [END]). If the predicted type is a building block or reaction token, a second-level classifier is used to predict the specific token within that class. This two-level classifier is similar to the routing mechanism in mixture-of-expert models (Eigen et al., 2013), as it avoids retrieving building blocks at every step. Formally, the conditional distribution of the next token is given by:

$$p(s_i|s_{<i}, \boldsymbol{C}) = \begin{cases} p\left(T(s_i) = \mathrm{BB} \mid \cdots\right) p\left(s_i \mid T(s_i) = \mathrm{BB}, \cdots\right) & T(s_i) = \mathrm{BB} \\ p\left(T(s_i) = \mathrm{RXN} \mid \cdots\right) p\left(s_i \mid T(s_i) = \mathrm{RXN}, \cdots\right) & T(s_i) = \mathrm{RXN} \ , \\ p\left(s_i \mid \cdots\right) & \text{otherwise} \end{cases} \tag{2}$$

where $T(s_i)$ denotes the type of the token $s_i$.

The transformer is trained using standard cross-entropy loss with causal masking to maximize the likelihood of the postfix notation sequence conditioned on their property prompts. At inference time, to generate molecules conditioned on a single property prompt, we first embed the prompt and prepend the embedding vectors to the [START] token. Then, we autoregressively sample tokens until the [END] token is emitted.

Note that we use a classifier to select building blocks from the library, with each class corresponding to a specific building block. This differs from previous models, which rely on molecular fingerprints to retrieve building blocks via deterministic nearest-neighbor retrieval (Luo et al., 2024). Fingerprint-based selection is limited to structural similarity and is not adaptable to varying contexts. In contrast, the classifier-based approach allows dynamic selection of building blocks based on the context provided by property prompts through the learnable unembedding matrix. In addition, it naturally allows probabilistic sampling from the building block library, which is crucial for generating diverse synthetic pathways in the property-conditioned setting as there is no one-to-one relationship between properties and molecules.

While direct tokenization of building blocks leads to the introduction of a large number of distinct tokens, the computational overhead is manageable at training time and negligible at inference time. The number of in-stock building blocks offered by commercial vendors today typically does not exceed one million. For example, as of November 2025, Enamine lists about 480 thousand building blocks in its global stock (Enamine, 2025) and WuXi lists about 90 thousand (WuXi AppTec, 2025). With an embedding dimension of 1,024

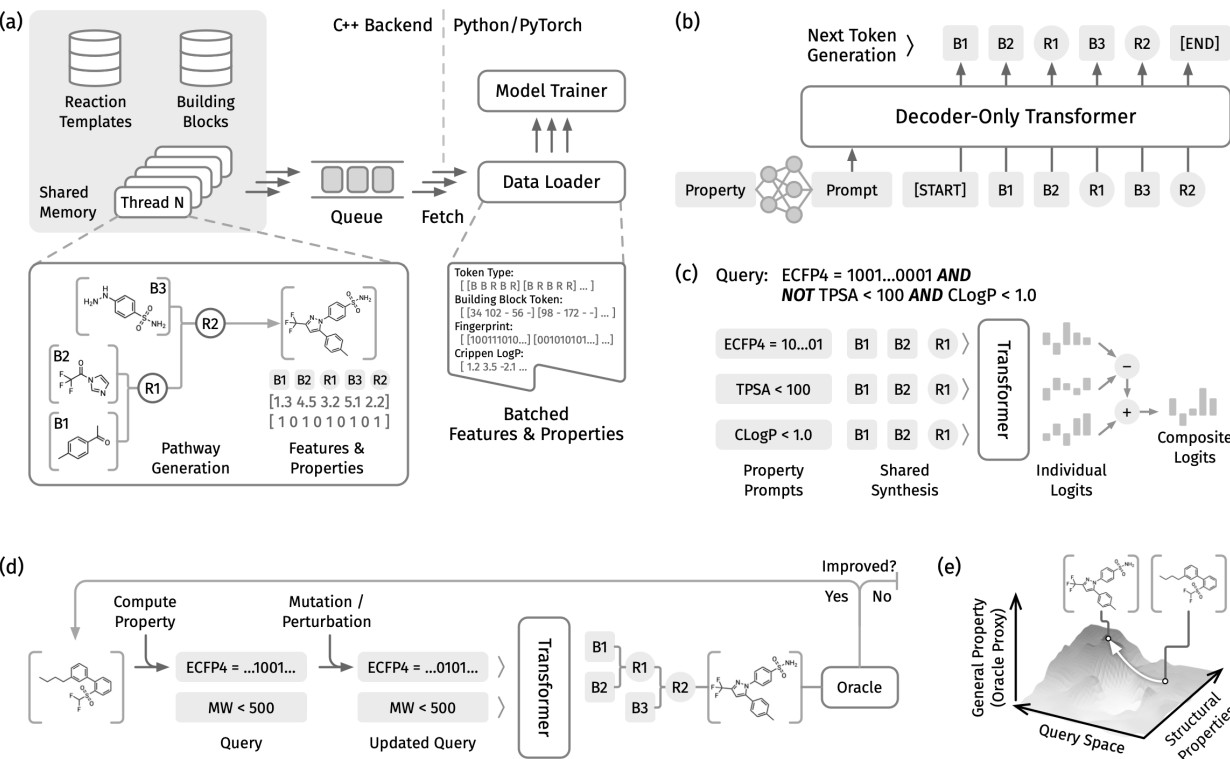

Figure 1: **(a)** The high-throughput C++ data engine generates synthetic pathways and computes molecular properties on the fly. It adopts a producer-consumer architecture, where multiple producer threads generate and featurize samples, which are then consumed by the Python-based training framework. **(b)** The decoder-only transformer architecture predicts the next token conditioned on property prompts. **(c)** Multiple properties in a composite query are used to condition the model separately. The resulting distributions are then combined. **(d)** Query space optimization. At each step, molecular properties are computed and perturbed to recondition the model. New molecules are then evaluated by oracle functions, and those with improved properties replace previous ones. **(e)** Since the structural properties used for training are sufficiently expressive to locate molecules, we can sample molecules with respect to general properties defined by black-box oracles by iteratively refining structural-property queries.

and 32-bit floating-point precision, the combined memory cost of the embedding and unembedding matrices for 1 million building blocks is $2 \times 10^6 \times 1,024 \times 4$ bytes $\approx 8$ GB, which is affordable on modern GPUs. If we reduce the floating point precision to 16-bit, the memory cost will be halved. At inference time, the embedding matrix does not need to be loaded into GPU memory, since only a small number of building block embeddings are looked up. Moreover, the unembedding matrix does not introduce significant inference-time memory overhead compared with fingerprint-based models (Luo et al., 2024; Gao et al., 2025), which store the all building block fingerprints in GPU memory for nearest-neighbor retrieval. Even if the number of in-stock building blocks grows drastically in the future, we can still use techniques such as sampled softmax (Blanc & Rendle, 2018) and cut cross-entropy (Wijmans et al., 2024) to keep the memory and computation bounded.

Another challenge with building block tokenization is that the embeddings are initialized randomly without any prior, which requires more training data than fingerprint-based representations encoding structural prior about building block. However, as we will describe in the next section, our high-throughput training data generation engine makes billion-scale training data feasible under moderate compute budget, thereby providing sufficient data for the model to learn meaningful building block embeddings from scratch.

## 2.2 High-throughput data engine enables billion-scale training datastream

Training data are generated on the fly by randomly sampling synthetic pathways from the combinatorial space defined by the building block library and reaction template set, paired with their molecular properties. This real-time approach enables virtually unlimited training data that scales with training time. However, its efficiency is limited in Python implementations (Luo et al., 2024) due to the lack of thread-level parallelism and the overhead of function calls through Python bindings, resulting in a bottleneck in training throughput given the highly optimized standard transformer architecture used.

To overcome this bottleneck, we developed a multi-threaded data pipeline in C++ that generates synthetic pathways and computes molecular properties using the RDKit C++ API (RDKit, 2010). It adopts a producer-consumer architecture (Figure 1a): multiple producer threads generate synthetic pathways, compute molecular properties, and featurize them into tensors, which are pushed into a queue buffer. The data loader of the training framework, which acts as the consumer, fetches featurized batches from the buffer, transfers them to GPU memory, and feeds them into the model for training.

The C++-based pipeline achieves a throughput of over 6,000 samples per second on 32 threads (16 cores) benchmarked on an AMD Ryzen Threadripper PRO 5975WX CPU, providing abundant throughput for training our transformer-based model on NVIDIA H100 GPUs. Another noteworthy advantage of our implementation is its low memory cost benefits from the shared memory space. It allows scaling up the number of threads without a significant increase in memory usage, whereas the previous Python-based implementation's memory cost grows linearly with the number of processes as each process requires a copy of the building block library. The model used for evaluation in this work was trained on two NVIDIA H100 GPUs with 32 CPU cores for 640,000 steps over 48 hours, using a batch size of 2,048 per step, resulting in a total of about 1.3 billion training pathways.

We also implemented a multi-threaded C++ function for batched detokenization at inference time. It takes as input a batch of postfix notation tokens, distributes the sequences to multiple threads, converts them into synthetic pathways, and obtains the product molecules with RDKit.

## 2.3 Training setup

Following previous works (Luo et al., 2024; Gao et al., 2025; Lee et al., 2025), we used Enamine US in-stock building block set retrieved on October 1, 2023 and the reaction template set curated by Gao et al. (2025) which contains 115 reaction templates.

The model consists of 12 transformer layers, with a model dimension of 1,024, a feedforward dimension of 2,048, and 16 attention heads. Excluding the embedding and unembedding matrices for building blocks, the transformer backbone together with other embedding layers has about 131 million parameters. When including these matrices, the total number of trainable parameters increases to 589 million. The model is optimized using the Adam optimizer with an initial learning rate of $3 \times 10^{-4}$ and a batch size of 2,048. A cosine annealing learning rate scheduler is used with $T_{\max} = 500$ and $\eta_{\min} = 10^{-5}$. The scheduler is triggered after each validation epoch. Validation is conducted every 2,000 training steps on random synthetic pathways. The model is trained using 32-bit full-precision floating point, with no mixed-precision or low-precision optimizations. Training runs for 48 hours on two NVIDIA H100 GPUs, corresponding to 640,000 iterations.

We train the model using the following structural properties widely used in quantitative structure-activity relationship (QSAR) studies (Roy et al., 2015): (1) ECFP4 fingerprint (Rogers & Hahn, 2010); (2) Gobbi pharmacophoric feature-based FCFP4 fingerprint (Gobbi & Poppinger, 1998); (3) BRICS decomposition-based substructures (Degen et al., 2008); and (4) physicochemical descriptors from RDKit including molecular weight, CLogP, TPSA, *etc* (RDKit, 2010).

Each training sample consists of only one property type randomly selected from the properties above paired with a synthetic pathway.

### 2.4 Generating molecules according to composite property queries

As the model is trained using pairs consisting of a single property and a synthetic pathway, it does not learn the distribution of molecules conditioned on multiple properties, which is often required in practical scenarios. A naive solution would be to include multiple properties in every training sample, but the amount of data required for training in this way grows exponentially with the number of property types (Du & Kaelbling, 2024). To avoid this bottleneck, we formulate a method to compose multiple properties at inference time. First, consider two different property prompts: $C_1$ and $C_2$; $p(s|C_1)$ and $p(s|C_2)$ are the distributions of postfix notations conditioned on each prompt respectively.

**AND (Conjunction)** The conditional distributions for molecules satisfying both conditions $C_1$ and $C_2$ is the product of the two distributions:

$$p(s|C_1 \wedge C_2) \propto p(s|C_1)^\alpha p(s|C_2)^\beta = \prod_{i=1}^{N} p(s_i|s_{<i}, C_1)^\alpha p(s_i|s_{<i}, C_2)^\beta \quad (\alpha, \beta > 0), \tag{3}$$

where $\alpha$ and $\beta$ are hyperparameters that control the relative importance of each condition. At each sampling step, we first compute the conditional distributions of the next token given each property prompt, and then combine them using the above equation to get the final distribution for sampling. Specifically, the combined distribution is given by:

$$p(s_i|s_{<i}, C_1)^\alpha p(s_i|s_{<i}, C_2)^\beta = \text{softmax}\left(\alpha z_1 + \beta z_2\right), \tag{4}$$

where $z_1$ and $z_2$ are the logits of the next token predicted by the model given property prompts $C_1$ and $C_2$ respectively.

**NOT (Negation)** Similarly, the conditional distribution for molecules satisfying $C_1$ but not $C_2$ is given by:

$$p(s|C_1 \neg C_2) \propto \frac{p(s|C_1)^\alpha}{p(s|C_2)^\beta} = \prod_{i=1}^{N} \frac{p(s_i|s_{<i}, C_1)^\alpha}{p(s_i|s_{<i}, C_2)^\beta} \quad (\alpha, \beta > 0). \tag{5}$$

The combined distribution for each sampling step is given by:

$$p(s_i|s_{<i}, C_1 \neg C_2) \propto \frac{p(s_i|s_{<i}, C_1)^\alpha}{p(s_i|s_{<i}, C_2)^\beta} = \text{softmax}\left(\alpha z_1 - \beta z_2\right). \tag{6}$$

Note that the negation operation can be unified with the conjunction operation by allowing $\beta$ in equation 3 to take negative values, which provides a convenient formulation for implementation.

**OR (Disjunction)** Under the disjunction of two conditions $C_1$ and $C_2$, the distribution of molecules satisfying either condition is given by:

$$p(s|C_1 \vee C_2) \propto \alpha p(s|C_1) + \beta p(s|C_2) = \alpha \prod_{i=1}^{N} p(s_i|s_{<i}, C_1) + \beta \prod_{i=1}^{N} p(s_i|s_{<i}, C_2). \tag{7}$$

Unlike the previous two cases, this distribution cannot be factorized autoregressively. Therefore, we sample full sequences from each conditional distribution separately and then merge the samples to obtain the final set of molecules.

**Composite logical queries** We define *query* ($Q$) as a logical expression over molecular properties composed using the logical operators AND, NOT, and OR. To enable sampling from composite logical queries involving multiple properties connected by logical operators, we first convert the query into disjunctive normal form (DNF) (Rosen, 2019), *i.e.* a series of ORs where each term only contains ANDs and NOTs. For each conjunctive term, we apply equations 4 and 6 to get its composed distribution, from which samples

are drawn. Finally, we merge the samples from all conjunctive terms to obtain the final set of molecules satisfying the composite logical query.

The theoretical assumption underlies the above formulation is that property prompts are mutually conditionally independent given the molecule, and that the prior distribution over molecules is uniform. Under these assumptions, the joint conditional distribution can be expressed in a product-of-experts form as shown in equations 3 and 5 (Hinton, 1999). Although some structural properties may be independent, many molecular properties are correlated, particularly those related to size such as molecular weight and number of rotatable bonds. Nevertheless, this simplifying assumption is what allows us to derive a practical algorithm for composing multiple properties. In practice, this assumption becomes problematic when there are contradictions among properties, such as simultaneously requiring molecular weight below 100 and number of heavy atoms above 50, which we consider detectable and avoidable in general.

## 2.5 Synthesizable molecule sampling in query space

Although the properties used for training already span a broad range of structural and physicochemical characteristics that are frequently used in cheminformatics workflows, they do not cover the full space of molecular properties. We need a mechanism to sample molecules with respect to properties that are not included during training, such as docking scores (Santos-Martins et al., 2021), antibiotic activity (Stokes et al., 2020), and so on. A straightforward way is to fine-tune the model on the new properties, but this is inefficient and might require a large amount of property evaluations for training.

Fortunately, the property-based querying capability of PrexSyn opens a new avenue for sampling molecules with respect to general properties. As the structural properties used for training are sufficiently expressive to locate molecules within the synthesizable chemical space, we can optimize molecules with respect to black-box oracle functions by iteratively refining property queries until satisfactory candidates are found (Figure 1e). This makes PrexSyn a general-purpose tool for synthesizable molecular design.

The sampling process (Figure 1d) begins with a seed query $Q_0$, which is used to generate an initial set of candidate molecules. At each iteration, each candidate molecule $\mathcal{M}$ is mapped to a property query $Q_t$. In general, the property query takes the form $Q = C_1^{(\mathrm{opt})}(\mathcal{M}) \wedge C_2^{(\mathrm{opt})}(\mathcal{M}) \wedge \cdots C_1^{(\mathrm{cstr})} \wedge \cdots$, where $\{C_i^{(\mathrm{opt})}(\mathcal{M})\}$ denotes optimizable conditions dependent on $\mathcal{M}$ and $\{C_i^{(\mathrm{cstr})}\}$ denotes fixed constraints. Next, perturbation is added to the optimizable term, resulting in an updated query $Q' = C_1'^{(\mathrm{opt})} \wedge \cdots C_1^{(\mathrm{cstr})} \wedge \cdots$. The updated query is then used to generate new candidate molecules, which are evaluated by the oracle. New candidates that achieve a better oracle score will replace the old candidates, leading to an improved set of molecules. Note that this process is described generically, and it may be implemented using genetic algorithms, Metropolis-Hastings sampling, or other iterative sampling methods. These implementations differ in how property queries are updated and how new candidates are selected.

# 3 Results

## 3.1 PrexSyn achieves state-of-the-art accuracy and speed in chemical space projection

We first evaluate PrexSyn on the chemical space projection task. This task involves finding synthesizable analogs for given molecular graphs. Two benchmark datasets with different emphases (Luo et al., 2024; Gao et al., 2025) are used for evaluation: (1) **Enamine testset**: 1,000 molecules curated from the Enamine REAL database (Shivanyuk et al., 2007), used to assess how well the model covers the synthesizable chemical space. (2) **ChEMBL testset**: 1,000 molecules from ChEMBL (Gaulton et al., 2012), which are not necessarily synthesizable with the Enamine building blocks and reactions, used to test the model's ability to find synthesizable analogs for arbitrary molecular graphs.

To run the projection task, we first compute the ECFP4 fingerprint of each molecule in the testset and embed it into prompt vectors, which condition the model to generate postfix notations of synthesis. 64, 128, and 256 independent samples are drawn for each target molecule. The product with the highest fingerprint similarity

Table 1: Chemical space projection results. *Recons.%* denotes the reconstruction rate, defined as the fraction of molecules with a similarity score of 1. *Similarity* denotes the Tanimoto similarity based on ECFP4 (Morgan) fingerprint. PrexSyn achieves both the highest accuracy and the highest efficiency.

| Method | Enamine REAL | | ChEMBL | | Time/Target |
| --- | --- | --- | --- | --- | --- |
| | Recons.% | Similarity | Recons.% | Similarity | |
| SynNet (Gao et al., 2021) | 11.0% | 0.57 | 5.4% | 0.43 | - |
| ChemProjector (Luo et al., 2024) | 46.0% | 0.81 | 13.0% | 0.60 | 5.15s±4.58s |
| SynthesisNet (Sun et al., 2024) | - | - | 9.2% | 0.53 | - |
| SynLlama (Sun et al., 2025) | 69.1% | 0.92 | 19.7% | 0.68 | 24.88s±13.24s |
| SynFormer (Gao et al., 2025) | 66.10% ±0.65% | 0.9137 ±.0014 | 20.67% ±0.72% | 0.6737 ±.0005 | 3.45s±3.60s |
| ReaSyn (Lee et al., 2025) | 74.93% ±0.17% | 0.9403 ±.0010 | 22.07% ±0.21% | 0.6740 ±.0007 | 19.71s±6.68s |
| PrexSyn (#samples=64) | 92.64% ±0.30% | 0.9819 ±.0005 | 25.40% ±0.31% | 0.7300 ±.0008 | **0.10s**±0.03s |
| PrexSyn (#samples=128) | 93.60% ±0.23% | 0.9845 ±.0004 | 27.18% ±0.33% | 0.7429 ±.0006 | 0.15s±0.04s |
| PrexSyn (#samples=256) | **94.06%** ±0.26% | **0.9859** ±.0007 | **28.32%** ±0.20% | **0.7533** ±.0011 | 0.26s±0.06s |

to the input molecule is selected as the projection result in accordance with the evaluation procedure of previous studies. All inference is conducted on a single NVIDIA 4090 GPU.

As shown in Table 1 and Figure 2, PrexSyn significantly surpasses previous methods on both datasets in terms of quality and efficiency. In particular, it achieves a record-high reconstruction rate of 94.06% on the Enamine testset, along with a Tanimoto similarity score over Morgan fingerprints of 0.9859, whereas the previous best model only achieves a reconstruction rate of 76.8% and a similarity score of 0.95. This result shows that PrexSyn nearly perfectly covers the synthesizable chemical space defined by the Enamine building blocks and reactions, thereby establishing a strong foundation for more complex chemical space exploration tasks. On the ChEMBL testset, PrexSyn achieves a reconstruction rate of 28.32% and a similarity score of 0.7533, setting a new state-of-the-art performance, which demonstrates its strong capability to find similar analogs for molecules beyond the defined chemical space.

In addition to the improved accuracy, PrexSyn is substantially faster, taking an average of only 0.10 seconds per target with a sample size of 64 and 0.26 seconds with a sample size of 256, which is at least 60 times faster than the highest-accuracy baseline. The high inference speed of PrexSyn is enabled by two factors: the C++-based detokenizer (Section 2.2) and the stronger model capacity that eliminates the need for inference-time search. While previous methods (Luo et al., 2024; Gao et al., 2025; Lee et al., 2025) rely on beam search or other inference-time scaling techniques to maximize the similarity scores, PrexSyn simply generates multiple postfix notations independently in parallel and convert all of them into synthetic pathways at once. This efficiency makes PrexSyn practical for large-scale applications.

We also trained variants of PrexSyn with different amounts of training data to evaluate the effect of data volume on model performance. As shown in Figure 2c, the reconstruction rate consistently increases as the training data scale grows, highlighting the benefit of large-scale training. Notably, training the full-scale PrexSyn on 1.3 billion samples requires only 48 hours on 2 NVIDIA H100 GPUs. This training cost is substantially lower than that of SynFormer (Gao et al., 2025) and ReaSyn (Lee et al., 2025). For comparison, SynFormer required 6 days on 8 A100 GPUs to train on 742 million samples, and ReaSyn required 5 days on 16 A100 GPUs to train on 512 million samples.

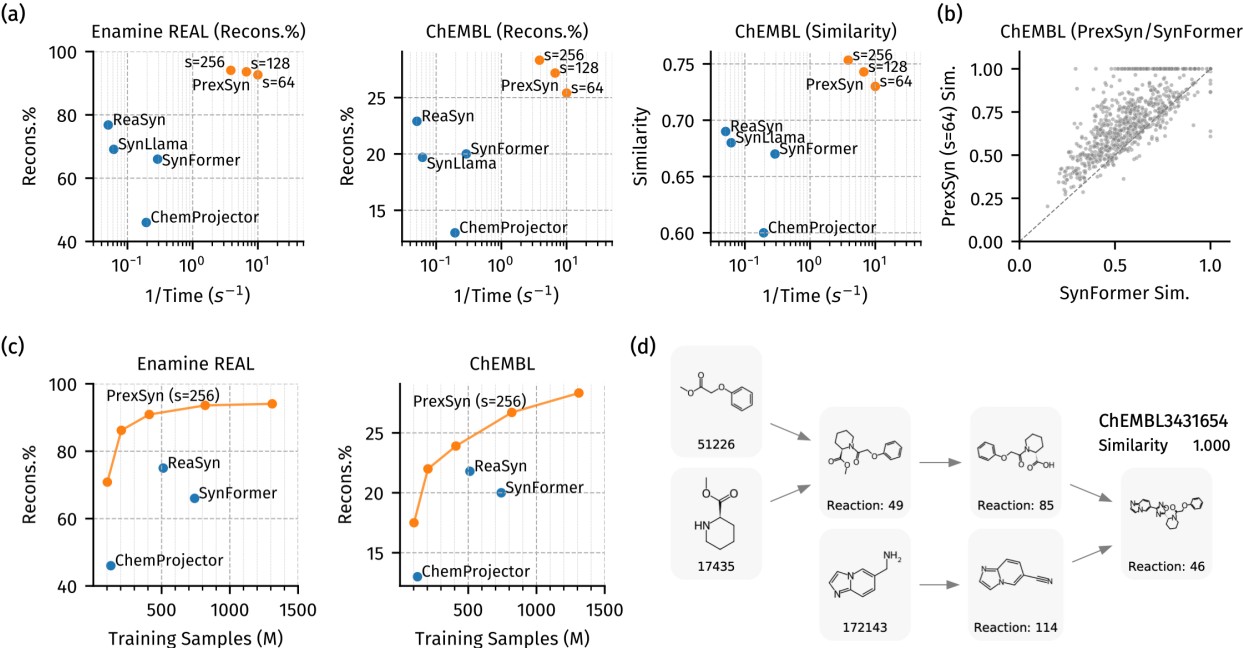

Figure 2: **(a)** Reconstruction rate/similarity versus inverse inference time on the Enamine and ChEMBL test sets. PrexSyn outperforms all baselines in both accuracy and efficiency. **(b)** Similarity of molecules projected by PrexSyn versus SynFormer on the ChEMBL test set. Each point corresponds to a molecule; points above the diagonal indicate higher similarity achieved by PrexSyn. The majority of points lie above the line, demonstrating consistently better performance. **(c)** Reconstruction rate versus training data scale. The reconstruction rate increases with larger training data, highlighting the benefit of large-scale training. **(d)** Example projection of a molecule from the ChEMBL test set that PrexSyn reconstructs perfectly. Tanimoto similarity using Morgan fingerprints is shown.

## 3.2 PrexSyn achieves state-of-the-art efficiency in synthesizable molecular sampling

To quantify the general sampling efficiency of PrexSyn, we conduct evaluation on seven multiproperty objectives and one rediscovery task from the GuacaMol benchmark suite (Brown et al., 2019). In these settings, scoring functions are treated as black boxes, which means no information about their internal form is visible to the model — only the final scores are provided. We set the initialization query of the $Q_0$ to Lipinski's Rule of 5 to generate drug-like molecules as starting candidates. The optimizable term $C^{(\text{opt})}$ is defined as the ECFP4 fingerprint and no constraint term is applied, $C^{(\text{cstr})} = \varnothing$. Genetic algorithm is used to optimize the fingerprint $C^{(\text{opt})}$ in alignment with previous methods (Gao et al., 2025; Jensen, 2019; Lee et al., 2025). The population size is set to 500, and at each iteration, 50 parents are sampled to generate 50 offspring through crossover.

We compare PrexSyn with both synthesis-agnostic and synthesis-based baselines. All methods are given a budget of 10,000 oracle calls, and AUC-Top10 scores are reported following previous studies (Gao et al., 2022). As shown in Table 2, PrexSyn achieves the highest average score on 6 out of 8 tasks, outperforming all synthesis-agnostic and synthesis-based baselines.

These results highlight the advantage of the query-space optimization paradigm enabled by PrexSyn. We attribute the high sampling efficiency of PrexSyn to two main factors: First, the high coverage of synthesizable chemical space of PrexSyn provides access to a much larger and more diverse set of feasible molecules, which increases the likelihood of finding high-scoring candidates.

Second, the property-based queries are numerical and therefore provide a well-structured optimization landscape that is easier to navigate. Moreover, even when a query does not correspond to any valid molecule,

Table 2: GuacaMol benchmark results measured by AUC-Top10 (Gao et al., 2022). DoG-Gen (Bradshaw et al., 2020) generates synthetic pathways but does not guarantee the validity of building blocks; therefore, its synthesizability is marked as neither yes nor no (◆). PrexSyn achieves the highest sampling efficiency on 6 out of 8 targets while maintaining synthesizability.

| Method | Syn. | Amlo. | Fexo. | Osim. | Peri. | Rano. | Sita. | Zale. | Cele. |
|---|---|---|---|---|---|---|---|---|---|
| REINVENT (Olivecrona et al., 2017) | ✗ | 0.635 | 0.784 | 0.837 | 0.537 | 0.760 | 0.021 | 0.358 | 0.713 |
| GraphGA (Jensen, 2019) | ✗ | 0.651 | 0.785 | 0.829 | 0.533 | 0.745 | 0.524 | 0.458 | 0.682 |
| MolGA (Tripp & Hernández-Lobato, 2023) | ✗ | 0.688 | 0.825 | 0.844 | 0.547 | 0.804 | **0.582** | 0.519 | 0.567 |
| DoG-Gen (Bradshaw et al., 2020) | ◆ | 0.537 | 0.697 | 0.776 | 0.475 | 0.712 | 0.048 | 0.123 | 0.466 |
| SynNet (Gao et al., 2021) | ✓ | 0.567 | 0.764 | 0.797 | 0.559 | 0.743 | 0.026 | 0.341 | 0.443 |
| SyntheMol (Swanson et al., 2024) | ✓ | 0.004 | 0.703 | 0.823 | 0.013 | 0.767 | 0.000 | 0.000 | 0.527 |
| SynthesisNet (Sun et al., 2024) | ✓ | 0.608 | 0.791 | 0.810 | 0.524 | 0.741 | 0.313 | **0.528** | 0.582 |
| SynFormer (Gao et al., 2025) | ✓ | 0.696 | 0.786 | 0.816 | 0.530 | 0.751 | 0.338 | 0.478 | 0.559 |
| ReaSyn (Lee et al., 2025) | ✓ | 0.678 | 0.788 | 0.820 | 0.560 | 0.742 | 0.342 | 0.492 | 0.754 |
| PrexSyn | ✓ | **0.781** ±.023 | **0.837** ±.013 | **0.855** ±.007 | **0.714** ±.010 | **0.807** ±.009 | 0.471 ±.030 | 0.504 ±.018 | **0.801** ±.005 |

the model can still generate molecules that approximate the desired properties, thereby smoothing the optimization landscape. This is in contrast to the discrete and sparse search spaces of synthetic trees, where the action space of building block selection is intractably large and valid pathways are sparse because not all combinations of building blocks and reaction templates are valid.

As shown in Figure 3a, PrexSyn successfully reconstructs the target molecule and discovers multiple high-scoring analogs in the Celecoxib Rediscovery benchmark, whereas methods that directly search over synthetic trees fail to do so: SynFlowNet reports top 10 average similarity of 0.48 (Cretu et al., 2024), and SyntheMol achieves an AUC-Top10 of only 0.527 (Table 2).

### 3.3 PrexSyn enables molecular generation with composite property queries

We next design three query tasks that simulate real-world drug discovery scenarios where composite property constraints are involved, as summarized in Table 3. For each task, we generate 1,000 postfix notations and score the product molecules according to how well they satisfy the specified query. We report the mean and standard deviation of both the average score and the diversity of the top 5% and top 10% highest-scoring molecules across 5 independent runs. Diversity is quantified as 1 minus the average pairwise Tanimoto similarity between Morgan fingerprints of the generated molecules (Jin et al., 2020).

**Task 1** evaluates the generation of drug-like molecules that satisfy Lipinski's Rule of Five (Lipinski, 2004; Chagas et al., 2018), expressed as a conjunction of multiple property constraints. Molecules are scored between 0 and 1 according to the fraction of conditions satisfied. The generated set achieves an average score of 0.9549, with the top 5% and 10% of molecules achieving perfect scores of 1.0000. These top samples also maintain high diversity (above 0.89), indicating that the model produces a broad and varied set of Lipinski-compliant molecules.

**Task 2** is inspired by the GuacaMol benchmarks (Brown et al., 2019), which involve finding analogs of existing drugs with modified physicochemical properties. Unlike the original GuacaMol benchmark, this task treats objectives as whitebox desiderata rather than blackbox oracles. Generated molecules are evaluated using the corresponding GuacaMol scoring functions, ranging from 0 to 1, with higher scores indicating better satisfaction of the desired properties. On the Cobimetinib optimization benchmark (Task 2), our best molecule achieves a score of 0.9326, with the top 5% and 10% averaging 0.8975 and 0.8848, respectively. For comparison, a graph-based approach that explicitly optimizes the four conditions, rather than treating the scoring function as a black box, reported a top score of 0.93 (Verhellen, 2022).

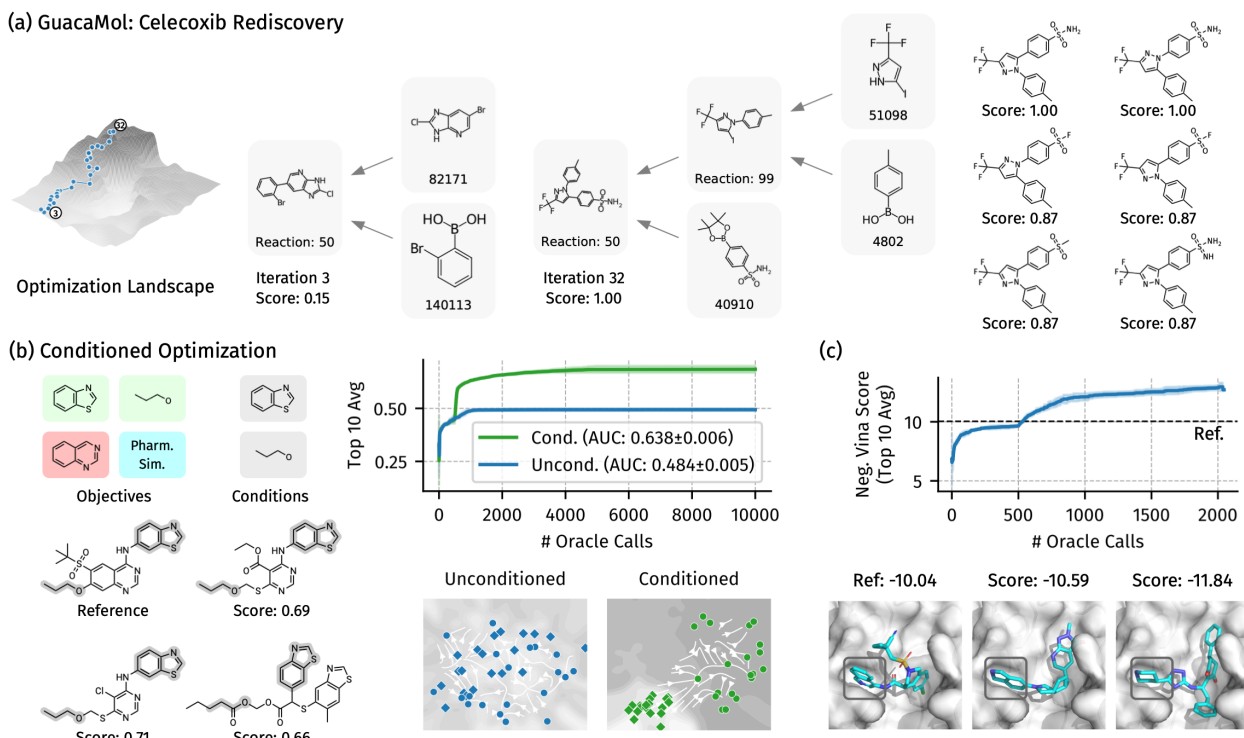

Figure 3: **(a)** Optimization trajectory, two snapshots, and six high-scoring samples for the Celecoxib Rediscovery task. PrexSyn successfully reconstructs Celecoxib and discovers high-scoring analogs. **(b)** Optimization trajectory on the scaffold hopping task with composite property queries. The composite query-conditioned optimization (green curve) achieves higher efficiency than the condition-free baseline (blue curve) and the optimization landscape is smoother. **(c)** Docking score optimization trajectory on the Mpro2 task. PrexSyn generates molecules that achieve improved docking scores compared to the baseline inhibitor (dashed line). The generated molecules share binding modes similar to the baseline inhibitor, including fitting into the highlighted subpocket.

**Task 3** is based on another GuacaMol benchmark that focuses on optimizing an Osimertinib MPO score. Our best molecule achieves a score of 0.9164, with the top 5% and 10% averaging 0.8314 and 0.8068, respectively. While the strongest baseline reported by Brown et al. (2019) achieved a slightly higher top score of 0.95, their method directly modifies molecular graphs without guaranteeing synthesizability.

**Composite query-based optimization** We further design a scaffold hopping task to demonstrate the optimization capability with respect to composite queries. Given a reference molecule, the goal of this task is to generate molecules that (1) have the same key substructures, (2) have a different scaffold, and (3) share similar pharmacophore features. These goals are wrapped into a scoring function consisting of three terms as the objective.

At each optimization iteration, the optimizable term $C^{(\text{opt})}$ of the query is defined as the ECFP4 fingerprint and the constraint term is to require the two key substructures, *i.e.* $C^{(\text{cstr})} = \texttt{Substruct}(D_1) \wedge \texttt{Substruct}(D_2)$. The initialization query is Lipinski's Rule of 5 to seed an initial set of random drug-like molecules. We also run a baseline that directly optimizes the fingerprint of the full molecule without the decoration conditions. As shown in Figure 3b, the composite query-conditioned optimization (green curve) achieves higher efficiency than the condition-free baseline (blue curve) and the optimization landscape is smoother. This result demonstrates the composite query's ability to narrow down the search space in scenarios where partial information is available, leading to more efficient optimization.

Table 3: Composite query-based molecular generation results. Three tasks reflecting drug discovery scenarios with composite property constraints are designed. For each task, we report the mean and standard deviation of both the average score and the diversity of the top 5% and top 10% highest-scoring molecules across 5 independent runs. A high score indicates a high degree of compliance to the requested property constraints.

| # | Task Description | Query | Best Score | Average Score (↑) | | | Diversity (↑) | |
|---|---|---|---|---|---|---|---|---|
| | | | | T5% | T10% | All | T5% | T10% |
| 1 | Generate molecules that satisfy Lipinski's Rule of 5. | `MW<500 AND Donors<5 AND Acceptors<10 AND RotatableBonds<10 AND TPSA<140 AND CLogP<5.0` | 1.0000 ±.0000 | 1.0000 ±.0000 | 1.0000 ±.0000 | 0.9549 ±.0036 | 0.8902 ±.0011 | 0.8902 ±.0011 |
| 2 | Find analogs of Cobimetinib that have 3 rotatable bonds and 3 aromatic rings. Crippen logP should not exceed 5.0 | `ECFP4("OC1(CN...") AND RotatableBonds=3 AND AromaticRings=3 AND CLogP<5.0` | 0.9326 ±.0040 | 0.8975 ±.0013 | 0.8848 ±.0017 | 0.7108 ±.0060 | 0.6017 ±.0113 | 0.6770 ±.0176 |
| 3 | Reduce the lipophilicity of Osimertinib by increasing TPSA to above 100 and reducing logP to below 1.0 | `ECFP4("COc1cc...") AND NOT TPSA<100 AND CLogP<1.0` | 0.9164 ±.0217 | 0.8314 ±.0081 | 0.8068 ±.0047 | 0.4971 ±.0085 | 0.7499 ±.0234 | 0.8024 ±.0061 |

## 3.4 PrexSyn is effective at molecular optimization using docking oracles

**sEH** We evaluate PrexSyn on the task of generating ligands for soluble epoxide hydrolase (sEH). The oracle function is defined as the negative docking score predicted by a proxy model trained on molecules docked with AutoDock Vina against the sEH protein structure (Cretu et al., 2024; Bengio et al., 2021). Following Cretu et al. (2024), the predicted scores are normalized by a factor of $1/8$.

The setting of the baseline method SynFlowNet (Cretu et al., 2024) differs from ours. SynFlowNet is trained to learn a distribution of molecules with good docking score, whereas we focus on directly optimizing docking score starting from random molecules. Once trained, SynFlowNet can generate molecules in a single forward pass but requires extensive training data and oracle calls (5000 steps, batch size 64, totaling ∼300k samples). In contrast, PrexSyn requires multiple optimization iterations but does not rely on any training samples. While the two settings are not directly comparable, we can still view PrexSyn as a sampler with warm-up steps and compare the quality of generated molecules under a stricter oracle budget. Specifically, we allow PrexSyn 10,000 oracle calls, about 30 times fewer than SynFlowNet, and evaluate performance using the top 1,000 generated molecules.

PrexSyn achieves a mean sEH score of 1.01, significantly outperforming SynFlowNet's best-reported score of 0.94 (Appendix Table 4). Note that since the score is defined as the negative binding energy divided by 8, values above 1.0 are possible. In addition, our generated molecules achieve better drug-likeness, with a QED score of 0.80 compared to SynFlowNet's 0.68, and an improved SA score of 2.23 versus SynFlowNet's 2.67. While we include the SA score (Ertl & Schuffenhauer, 2009) in the comparison for completeness, we note that SA scores are less relevant in the context of synthesizable molecular design, as providing a synthetic pathway composed only of purchasable building blocks and reaction templates, which both methods do, is a much stronger evidence of synthesizability than the heuristic SA score.

**Mpro2** We further evaluate PrexSyn on the task of generating ligand candidates for the SARS-CoV-2 main protease (Mpro2) using AutoDock-GPU (Santos-Martins et al., 2021) as the oracle function. For this task, we use the protein structure from PDB entry 7GAW, where an inhibitor discovered through the COVID Moonshot project (Boby et al., 2023) is co-crystallized with Mpro2. This inhibitor serves as the baseline molecule for evaluation. We set the initialization query to Lipinski's Rule of 5 to generate drug-like molecules as starting candidates and perform genetic algorithm over the query space containing the ECFP4 fingerprint as the optimizable term. 2,000 oracle calls are budgeted. As shown in Figure 3c, PrexSyn can generate molecules from scratch that achieve improved docking scores compared to the baseline inhibitor.

Visualizations of the docking poses further reveal that the generated molecules share binding modes similar to the baseline inhibitor, including fitting into the highlighted subpocket.

## 4 Conclusion

PrexSyn pushes the frontier of synthesizable molecular design through its state-of-the-art efficiency and programmability. Its efficiency is reflected in three key aspects: first, an efficient high-throughput data engine that enables billion-scale training; second, near-perfect coverage of synthesizable chemical space with state-of-the-art accuracy and speed; and third, state-of-the-art sampling efficiency for black-box oracle functions via query space optimization. Its programmability is manifested in two main features: first, support for composite property queries joined by logical operators, allowing users to "program" generation objectives; and second, the ability to optimize molecules with respect to black-box oracle functions based on this querying capability. Together, these capabilities make PrexSyn a powerful tool for molecular design and optimization, highlighting its potential to further the practical impact of generative AI.

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

## A  Related Work

Synthesizable molecular design methods can be roughly divided into two categories. The first directly searches the combinatorial space of synthetic pathways (Vinkers et al., 2003; Hartenfeller et al., 2012; Korovina et al., 2020; Gottipati et al., 2020; Horwood & Noutahi, 2020; Bradshaw et al., 2020; Gao et al., 2021; Swanson et al., 2024; Cretu et al., 2024; Seo et al., 2024; Koziarski et al., 2024; Zhu et al., 2025). Their performance remains limited due to several unaddressed challenges. First, the action space of building block selection is extremely large, the size of which often reaches hundreds of thousands or millions (Enamine, 2025). They generate synthetic pathways by explicitly selecting building blocks from such large libraries, which is difficult to sufficiently explore chemical space during both training and sampling. Second, the search space of synthetic pathways is sparse as not all combinations of building blocks and reactions lead to valid syntheses. Lastly, the representation of synthetic pathways is not informed of the structural features of the resulting product molecules. Models that learn distributions over synthetic pathways are typically only aware of the combinatorial rules of reactions and building blocks. As a result, the structural features of the products, which are crucial determinants of molecular properties, are not captured, leading to a gap between pathway generation and property optimization. These issues limit the sampling efficiency of this category of methods, making them even less practical for real-world applications that require expensive property evaluations or involve multiple property constraints.

The second category trains models to construct synthetic pathways from input molecular graphs (Luo et al., 2024; Gao et al., 2025; Sun et al., 2024; 2025; Lee et al., 2025). These models, however, cannot generate molecules conditioned on property specifications and remain limited in both chemical space coverage and efficiency. Beyond these two categories, other methods focus on specific problem classes, such as structure-based drug design (Jocys et al., 2024; Rekesh et al., 2025), or formulate the task differently, for instance, treating synthesizability as an optimization objective (Guo & Schwaller, 2024; 2025).

## B  Additional Results

Table 4: sEH docking-based molecular design results. PrexSyn outperforms all the baselines in sEH and QED scores.

| Method | Syn. | sEH($\uparrow$) | SA($\downarrow$) | QED($\uparrow$) |
|---|---|---|---|---|
| FragGFN (Bengio et al., 2021) | ✗ | 0.77±0.01 | 6.28±0.02 | 0.30±0.01 |
| FragGFN(SA) (Bengio et al., 2021) | ✗ | 0.70±0.01 | 5.45±0.05 | 0.29±0.01 |
| SyntheMol (Swanson et al., 2024) | ✓ | 0.64±0.01 | 3.08±0.01 | 0.63±0.01 |
| SynFlowNet (Cretu et al., 2024) | ✓ | 0.92±0.01 | 2.92±0.01 | 0.59±0.02 |
| SynFlowNet(SA) (Cretu et al., 2024) | ✓ | 0.94±0.01 | 2.67±0.03 | 0.68±0.01 |
| SynFlowNet(QED) (Cretu et al., 2024) | ✓ | 0.86±0.03 | 4.02±0.26 | 0.74±0.04 |
| ReaSyn (Lee et al., 2025) | ✓ | 0.97±0.01 | **2.01±0.02** | 0.75±0.01 |
| PrexSyn | ✓ | **1.01±0.00** | 2.23±0.04 | **0.80±0.01** |

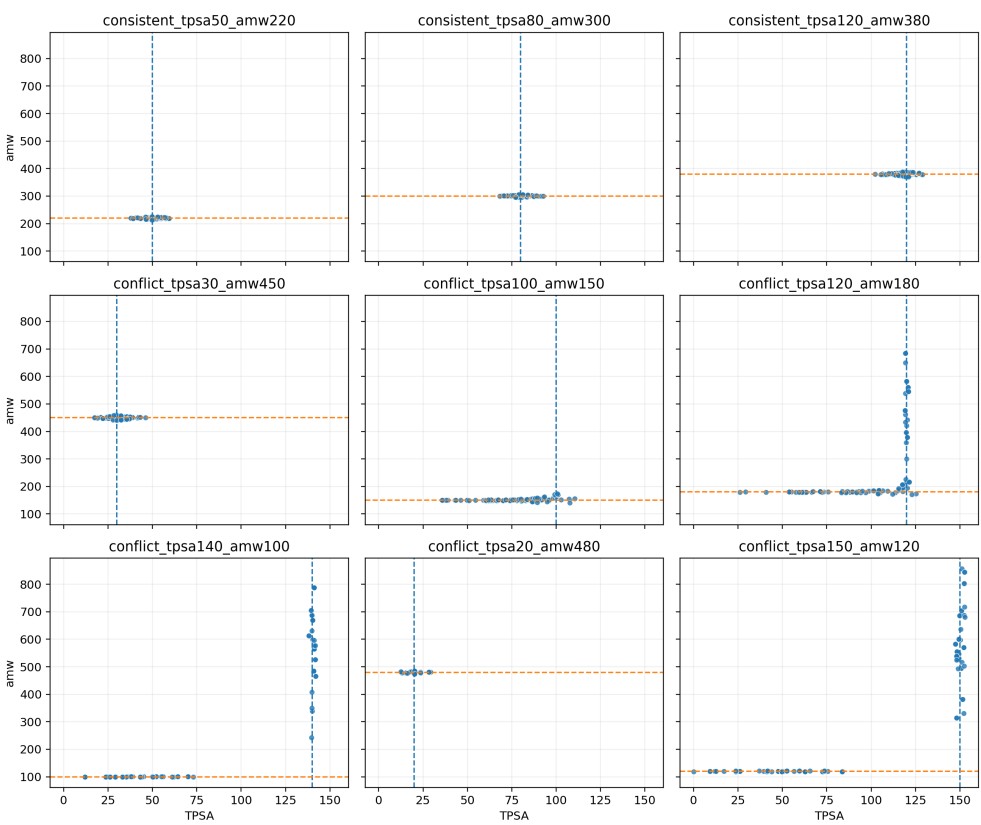

Figure 4: Generation results under consistent and conflicting combinations of TPSA and AMW conditions. Dashed lines indicate the target values for each property. When two conditions are in conflict (e.g., TPSA=150 and AMW=120), the model tries to satisfy at least one of the conditions.

