# OpenReview forum: "Efficient and Programmable Exploration of Synthesizable Chemical Space"
_TMLR — Accepted by TMLR_

### Review · Reviewer_Cd7n · 2026-03-11

**Summary Of Contributions:**

This paper proposes PrexSyn: a large-scale transformer that generates a synthesis tree given property constraints. Authors show how logical combinations of several constraints (e.g. AND) translate to algebraic operations on model's probability distributions, which enables more complex queries. Allowing for automated mutation of the property constraints yields a molecular optimization algorithm, which is shown to perform very competitively, even compared to synthesis-agnostic baselines that don't try to control for synthesizability.

**Audience:**

Yes

**Audience Explanation:**

PrexSyn appears to be a practically useful method for drug discovery, and would be an interesting read for anyone working in that space.

**Broader Impact Concerns:**

I don't see particular impact concerns with this work.

**Claims And Evidence:**

Yes

**Claims Explanation:**

The paper claims that PrexSyn can very efficiently choose synthesizable analogs, and optimize arbitrary properties.

For synthesizable analog generation, experiments show near perfect coverage of Enamine's space, with almost negligible sampling time; coverage of ChEMBL leaves room for improvement but it is still better than previous methods.

Results on GuacaMol show good optimization performance for arbitrary black-box properties. However, I'm not sure about the values reported for baselines: I recall that well-performing methods on GuacaMol can get average score over the tasks of 0.9 or higher, whereas this manuscript reports most individual task results around 0.6-0.8.

Further experiments on white box scoring functions, scaffold hopping, and docking, show PrexSyn works well across a variety of settings.

PrexSyn is trained on a very large data volume, and authors ablate the training set size, showing strong improvements going even beyond 0.5B training samples.

Finally, it is notable that the work puts a lot of stress on efficient implementation, and shows evidence that the efforts to implement things carefully pay off.

**Requested Changes:**

I would ask the authors to clarify the Guacamol results of the baseline methods.

---

> ### Author Response · Authors · 2026-03-28
>
> Thank you for the helpful comment!
>
> The GuacaMol scores (0.6–0.8) reported in our paper use the AUC-Top10 metric rather than the Top-X averages commonly used in earlier work.
>
> AUC-Top10, introduced by Gao et al. [1], measures the area under the curve of the average score of the top 10 molecules throughout the optimization process, thus capturing both final performance and sample efficiency. Because it averages performance over the entire trajectory, including early low-scoring stages, it is naturally lower.
>
> In contrast, Top-X metrics (e.g., average of top 10 at the end) only reflect the final outcome and do not penalize slow search, which is why they often exceed 0.9.
>
> Importantly, our reported results for synthesis-agnostic baselines are consistent with those reported in the benchmark study by Gao et al. [1].
>
> [1] Gao, W., Fu, T., Sun, J. and Coley, C., 2022. Sample efficiency matters: a benchmark for practical molecular optimization. Advances in neural information processing systems, 35, pp.21342-21357.

---

### Review · Reviewer_WnT5 · 2026-03-20

**Summary Of Contributions:**

This work presents PrexSyn, a generative model that:
(1) guarantees synthesizability according to reaction rules
(2) has high coverage of the synthesizable coverage space
(3) has high inference speed
(4) has high sample efficiency.

PrexSyn is trained using a two tier-approach, where the first tier predicts the type of the next token, and the second-tier predicts the specific token in the class chosen by the first tier. This two-tier approach helps PrexSyn avoid retrieving building blocks at each step.


The authors have proposed a multi-threaded data engine using the RDKit C++ API enabling training on 1.3 billion synthetic pathways in 48 hours.

PrexSyn formulates multiple molecular properties at inference time using a product-of-experts approach, avoiding the exponential data requirements of multi-condition training.

**Audience:**

Yes

**Audience Explanation:**

Yes, the paper is relevant to TMLR. The paper tackles the synthesizability gap by proposing two fundamental solutions - improving the systems framework by using a multi-threaded C++ engine and using probabilistic logic at inference.

**Broader Impact Concerns:**

A broader impacts section is not provided and not required.

**Claims And Evidence:**

Yes

**Claims Explanation:**

1. The authors have demonstrated an experimental analysis to highlight the improved speed and accuracy of PrexSyn as compared to other baseline approaches. The authors attribute this increased speed to the C++ based detokenizer and the increased capacity of the model. I would be curious to know how much does each factor contribute to the increased inference speed.

2. The authors have conducted experiments to showcase the efficiency of PrexSyn in sampling synthesizable molecules.

3. PrexSyn outperforms synthesis-agnostic and synthesis-based models on 6/8 GuacaMol Benchmark tasks.

4. Unlike the graph based models, PrexSyn outputs postfix notation tokens that correspond directly to sequences of chemical reactions and building blocks. This helps PrexSyn achieve greater construction synthesizability.

**Requested Changes:**

1. The authors have mentioned the use of classifier to select building blocks from the library. How does the approach scale to the number of building blocks. An ablation study might help the reader understand the performance against the size of the classifier output and how it scales to different number of outputs.

2. The authors talk about the use of oracle functions to evaluate the new molecules. How are these oracle functions defined? How do the authors ensure that the oracle functions cover the entire space of desirable and undesirable traits and qualities of the new molecules? Can the authors add more information about the oracle functions?

3. How did the authors determine the alpha and beta hyperparameters that control the relative importance of each condition? How does it scale to having more than two conditions? Can the authors elaborate on the choice of the hyperparameters? What happens when two conditions are related to each other?

4. While the results demonstrate that PrexSyn is faster than existing baselines, it would be nice to have an ablation study that compares how much of the speed up is enabled by the C++ based detokenizer as compared to having a stronger model.

5. It would be nice to have the high-throughput data generation engine on easily accessible websites like github if possible.

---

> ### Author Response · Authors · 2026-04-05
> **Response (1/2)**
>
> Thank you for the thoughtful review and for recognizing the contributions of PrexSyn in addressing the synthesizability gap. Below are our responses to the questions:
>
> ---
>
> **[1] Scalability of building block classifier**
>
> To address this, we have conducted an empirical ablation study on the parameter count and memory scaling behavior as the number of building blocks and batch sizes vary. The results are summarized in the table below:
>
>
> | **num_bb** | **batch size** | **#params** | **model size (MiB)** | **fwd peak memory (MiB)** | **fwd+bwd peak memory (MiB)** | **bb_embedding num parameters** | **bb_embedding size (MiB)** | **bb_classifier #params** | **bb_classifier size (MiB)** |
> | - | - | - | - | - | - | - | - | - | - |
> | **50K** | 1 | 210.8M | 804.30 | 815.52 | 1836.13 | 51.2M | 195.31 | 51.2M | 195.50 |
> | **50K** | 128 | 210.8M | 804.30 | 1719.87 | 3409.63 | 51.2M | 195.31 | 51.2M | 195.50 |
> | **50K** | 1024 | 210.8M | 804.30 | 2360.64 | 14579.77 | 51.2M | 195.31 | 51.2M | 195.50 |
> | **100K** | 1 | 313.3M | 1195.12 | 2824.57 | 4031.38 | 102.4M | 390.62 | 102.5M | 391.01 |
> | **100K** | 128 | 313.3M | 1195.12 | 2498.75 | 4373.33 | 102.4M | 390.62 | 102.5M | 391.01 |
> | **100K** | 1024 | 313.3M | 1195.12 | 3138.02 | 15406.18 | 102.4M | 390.62 | 102.5M | 391.01 |
> | **200K** | 1 | 518.2M | 1976.75 | 4386.58 | 6376.53 | 204.8M | 781.25 | 205M | 782.01 |
> | **200K** | 128 | 518.2M | 1976.75 | 4065.02 | 6345.85 | 204.8M | 781.25 | 205M | 782.01 |
> | **200K** | 1024 | 518.2M | 1976.75 | 4713.29 | 17364.33 | 204.8M | 781.25 | 205M | 782.01 |
> | **500K** | 1 | 1.1B | 4321.64 | 8297.96 | 12633.14 | 512M | 1953.12 | 512.5M | 1955.03 |
> | **500K** | 128 | 1.1B | 4321.64 | 8755.49 | 12201.06 | 512M | 1953.12 | 512.5M | 1955.03 |
> | **500K** | 1024 | 1.1B | 4321.64 | 9397.26 | 23227.10 | 512M | 1953.12 | 512.5M | 1955.03 |
>
>
> Based on this data, we highlight two main takeaways regarding scalability:
>
> 1. **Linear scaling of parameters is not a practical bottleneck**: As expected, the number of parameters and the memory footprints of the building block embedder and classifier scale linearly with the number of building blocks. For instance, scaling the vocabulary from 50K to 500K increases the classifier size from 51.2 million (\~195 MiB) to 512.5 million parameters (\~1,955 MiB). However, as discussed in Section 2.1, the number of in-stock building blocks offered by commercial vendors today typically does not exceed one million. Because the commercial building block space is practically bounded, this linear scaling is easily accommodated by modern GPU memory capacities. Even if in the future the number of building blocks grows significantly, we can apply techniques such as mixed precision, quantization, and sampled softmax as discussed in the manuscript.
>
> 2. **Sublinear scaling with batch size**: Notably, the table demonstrates that both the inference (forward pass) and training (forward + backward pass) peak memory costs scale strictly sublinearly with respect to the batch size. For example, using a 500K building block library, increasing the batch size by over $1000\times$ (from 1 to 1024) only increases the forward peak memory marginally from 8,297.96 MiB to 9,397.26 MiB. This sublinear memory footprint ensures that PrexSyn can maintain high-throughput batched generation and training efficiency.
>
> ---
>
> **[2] Definition of oracle functions**
>
> In our framework, we treat scoring functions strictly as black boxes, meaning PrexSyn has no internal visibility into how they are computed. PrexSyn relies only on the final returned scores to navigate the optimization landscape.
>
> The specific oracles we utilized are defined by established benchmarks, such as GuacaMol and Autodock.
>
> Because we treat the oracles as black boxes, PrexSyn itself does not guarantee that these functions cover the entire space of desirable traits. However, PrexSyn can efficiently sample and optimize molecules according to the specific traits that the given oracle measures.

---

> ### Author Response · Authors · 2026-04-05
> **Response (2/2)**
>
> **[3] Hyperparameters $\alpha$ and $\beta$ and correlated conditions**
>
> In our implementation, we simply set all coefficients to 1. This straightforward approach naturally scales to more than two conditions by setting $\alpha_i = 1$ for each condition $i$.
>
> Regarding related or interacting conditions: our formulation relies on the simplifying theoretical assumption that property prompts are mutually conditionally independent. While many structural properties are inherently correlated, we found that setting all weights to 1 works robustly in practice for guiding the generation process.
>
> Regarding the question on what happens when conditions contradict each other, we conducted further experiments to evaluate the model's behavior. We specifically analyzed combinations of Topological Polar Surface Area (TPSA) and Average Molecular Weight (AMW).
> As shown in the additional scatter plot (updated manuscript Page 18 Figure 4), when two conditions are in conflict (e.g., TPSA=150 and AMW=120), the model tries to satisfy at least one of the conditions. These results demonstrate that the model manages correlated and even conflicting conditions reasonably well.
>
> ---
>
>
> **[4] Speed up enable by the detokenizer**
>
> To decouple the speedup of our C++ detokenizer from the model's efficiency, we conducted an ablation study breaking down inference into Model Time and Detokenization Time.
>
> We benchmarked ChemProjector's inference efficiency on 100 molecules from its test set. In ChemProjector, detokenization accounts for ~78% of the total inference time (4.05s). SynFormer and ReaSyn used ChemProjector's detokenizer implementation, so we estimate that they also spend a similar proportion of time on detokenization.
>
> In contrast, PrexSyn's C++ detokenizer **reduces the detokenization overhead to a negligible 0.01s-0.02s**.
>
> | **Model**                | **Total Time** | **Model Time** | **Detokenization Time** |
> | ------------------------ | -------------- | -------------- | ----------------------- |
> | **ChemProjector**        | 5.17s ± 4.58s  | 1.12s ± 0.62s  | 4.05s ± 3.58s           |
> | **PrexSyn** (sample=64)  | 0.10s ± 0.03s  | 0.09s          | 0.01s                   |
> | **PrexSyn** (sample=128) | 0.15s ± 0.04s  | 0.14s          | 0.01s                   |
> | **PrexSyn** (sample=256) | 0.26s ± 0.06s  | 0.24s          | 0.02s                   |
>
> ---
>
> **[5] Open source of PrexSyn Engine**
>
> We will open-source both PrexSyn and the C++-based PrexSyn Engine, along with the data, model checkpoints, and scripts.

---

### Review · Reviewer_Nf1k · 2026-04-01

**Summary Of Contributions:**

There are two major strength of the paper worth emphasizing...

First, what I like is that this isn’t just one idea—it’s somewhat clean integration of three things that actually work together: (1) a property-conditioned decoder-only model operating directly over synthesis pathways, (2) a high-throughput C++ data engine that makes billion-scale on-the-fly training practical, and (3) an inference-time query/composition mechanism that supports logical constraints and black-box optimization.

Though Individually none of these are entirely new, but the way they’re put together here is cohesive and feels genuinely novel at the system level.

The empirical gains are also hard to hand-wave away. The jump to ~94% reconstruction on Enamine REAL (vs ~75% prior SOTA) is pretty significant, and they also improve ChEMBL similarity while being much faster at inference

Secondly, Another thing I like is that authors don’t just optimize for one benchmark. They actually evaluate across multiple axes...chemical space projection, GuacaMol black-box optimization, composite query generation, and docking-based optimization. That gives a much more convincing picture that the system is both efficient and flexible, not just overfitted to a single setup.

**Audience:**

Yes

**Audience Explanation:**

it should appeal to people working on generative modeling, scientific ML (especially molecular design), and increasingly the “systems + modeling” intersection (large-scale data generation, efficient training pipelines, etc.). The idea of combining property-conditioned generation with programmable query composition and synthesizability constraints is also aligned with broader trends in controllable generation and structured reasoning in generative models.

**Claims And Evidence:**

Yes

**Claims Explanation:**

Please see my response to the major contributions.

**Requested Changes:**

Overall I’m leaning accept, but there are a few gaps that should be addressed to make the paper fully convincing:

1. Clarify and stress-test the compositional query assumption. The whole AND/NOT composition relies on conditional independence between properties, which is clearly not true in many cases. Right now this is acknowledged but not really evaluated.
I’d like to see:
a. Experiments on correlated properties (e.g., MW vs rotatable bonds)
b. Failure cases or degradation curves as constraints become more conflicting
c. Sensitivity to α / β weights

Without this, the “programmable” claim feels more heuristic, rather than a principled design.

2. Stronger ablations to isolate what actually works and not working.
The gains are quite significant, but it’s unclear to me what really drives them. Please clarify:

a. Data scale vs architecture
b. Classifier-based building block selection vs retrieval
c. Property-conditioned generation vs projection-based methods

3. Make comparisons more apples-to-apples (especially optimization tasks)
Some comparisons (e.g., sEH) are not directly equivalent settings. That’s fine to me, but might not pass the scrutiny of subject matter experts:

Either add baselines under the same budget/setup
Or clearly separate “indicative” vs “fair” comparisons

4. Better characterization of compositional query scalabilit. The current tasks are somewhat “clean.” I’d like to see (at least some discussion if too much effort is required):
a. Scaling to more constraints (3–5+ properties)
b. Near-infeasible or contradictory queries
c. Trade-off curves (constraint satisfaction vs diversity / validity)

5. Clarify the notion of synthesizability vs real-world practicality
The guarantee is valid under templates + building blocks, but this is not the same as real experimental feasibility.
A short discussion clarifying this distinction would help avoid overclaiming.

---

> ### Author Response · Authors · 2026-04-06
>
> Thank you for your constructive and encouraging review. Below are our responses to the questions:
>
> ---
>
> **[1] Compositional query assumption**
>
> We acknowledge that the conditional independence assumption is a mathematical relaxation used to make the inference algorithm tractable.
> We have added a stress-test in the updated appendix (Page 18, Figure 4) evaluating the model's behavior under various Topological Polar Surface Area (TPSA) and Average Molecular Weight (AMW) constraints. Our empirical results demonstrate that the model manages correlated and even conflicting conditions reasonably well.
>
> Because polar atoms contribute significantly to molecular mass, these properties are naturally correlated. Our results reveal that:
>
> - When conditions are aligned (e.g., TPSA=50, AMW=220), generated molecules cluster at the intersection of target values.
> - When conditions are in conflict (e.g., TPSA=150, AMW=120—an impossible combination as the necessary oxygen/nitrogen atoms to achieve that TPSA would exceed the weight limit), the model does not collapse to non-sensical outputs. Instead, it generates an L-shaped distribution, attempting to satisfy at least one of the conditions.
>
> ---
>
>
> **[2] What actually works**
>
> Figure 2(c) shows that at same training data scales (e.g., 500M or 750M samples), PrexSyn outperforms baselines like ReaSyn and SynFormer. This indicates that architectural improvements are the first contributor to performance gains. The training data engine enables scaling to larger data sizes with less time and compute, further amplifying the performance improvements.
>
> A core limitation of projection-based models is their reliance on deterministic fingerprint retrieval for building block selection. Fingerprint distances are strictly limited to static structural similarity and cannot adapt to varying generation contexts. Therefore, given that PrexSyn and previous models all use transformer-based architectures, the performance gains of PrexSyn can be attributed to the building block classifier, which enables dynamic, context-aware building block selection.
>
> Projection is a special case with in PrexSyn's property-conditioned framework. When the input is only the target molecule fingerprint, it is equivalent to projection.
>
> ---
>
>
> **[3] Make comparisons more apples-to-apples**
>
> We acknowledged that the sEH optimization setting for the baseline method (SynFlowNet) differs from ours. This is because the fundamental different in problem setting: SynFlowNet is trained to learn a distribution of molecules with good docking scores for one specific target, whereas PrexSyn focuses on optimizing any docking scores starting from random molecules.
>
> It is hard to make direct apples-to-apples comparisons, and therefore we agree that our comparison is "indicative". It is also worth noting that we allowed PrexSyn 10,000 oracle calls, about 30 times fewer than SynFlowNet.
>
> ---
>
> **[4] Compositional query**
>
> As discussed in our response to Question 1, our empirical results in Figure 4 show that the model manages correlated and even conflicting conditions reasonably well.
>
> As we add more conditions, the property landscape becomes more constrained. Therefore, we anticipate that the model will still try to satisfy at least one of the conditions, even if they are in conflict, but clearly the generation will become more challenging and our model might not be able to satisfy all conditions simultaneously.
>
>
> ---
>
> **[5] Synthesizability vs real-world practicality**
>
> We agree with the reviewer that logical "synthesizability" is not strictly equivalent to real-world practicality.
> In this work, as in previous works, we define synthesizability as the ability to construct a valid synthetic route using only given building blocks and predefined reaction templates.
> However, as noted in studies of enumerative libraries (e.g., Grygorenko et al., 2020), the logical validity of a virtual route does not guarantee its practicality in laboratory settings.
> Synthesis success rates vary depending on many factors including the building blocks and reactions used, with Grygorenko et al. estimating an overall success rate of around 85%.
> Therefore, we emphasize that our definition of logical synthesizability is a necessary, but not completely sufficient, condition for real-world practicality. Nevertheless, while the success rate is not perfect, ensuring logical synthesizability represents a non-trivial advancement compared to generating structurally unconstrained molecules without explicit consideration for synthesizability.

---

### Decision · Action_Editor_xqr9 · 2026-05-13

**Recommendation:** Accept as is

**Audience:**

Yes

**Audience Explanation:**

Effective and efficient methods to explore the ever-growing space of accessible chemistry are important, this paper will be relevant to the TMLR audience.

**Claims And Evidence:**

Yes

**Claims Explanation:**

The paper's empirical results clearly support the thesis.